# Collaborative Filtering with A Synthetic Feedback Loop

## Abstract

We propose a novel learning framework for recommendation systems, assisting collaborative filtering with a synthetic feedback loop. The proposed framework consists of a "recommender" and a "virtual user." The recommender is formulizd as a collaborative-filtering method, recommending items according to observed user behavior. The virtual user estimates rewards from the recommended items and generates the influence of the rewards on observed user behavior. The recommender connected with the virtual user constructs a closed loop, that recommends users with items and imitates the unobserved feedback of the users to the recommended items. The synthetic feedback is used to augment observed user behavior and improve recommendation results. Such a model can be interpreted as the inverse reinforcement learning, which can be learned effectively via rollout (simulation). Experimental results show that the proposed framework is able to boost the performance of existing collaborative filtering methods on multiple datasets.

## 1 Introduction

Recommendation systems are important modules for abundant online applications, helping users explore items of potential interest. As one of the most effective approaches, collaborative filtering Sarwar et al. (2001); Koren & Bell (2015); He et al. (2017) and its deep neural networks based variants He et al. (2017); Wu et al. (2016); Liang et al. (2018); Li & She (2017); Yang et al. (2017); Wang et al. (2018) have been widely studied. These methods leverage patterns across similar users and items, predicting user preferences and demonstrating encouraging results in recommendation tasks Bennett & Lanning (2007); Hu et al. (2008); Schedl (2016). Among these work, beside "user-item" pairs, side information, $e.g.$, user reviews and scores on items, are involved and have achieved remarkable success Menon et al. (2011); Fang & Si (2011). Such side information is a kind of user feedback to the recommended items, which is promising to improve the recommendation systems.

Unfortunately, both the user-item pairs and user feedback are extremely sparse compared with the search space of items. What is worse, when the recommendation systems are trained on static observations, the feedback is unavailable until it is deployed in real-world applications — in both training and validation phases, the target systems have no access to any feedback because no one has observed the recommended items. Therefore, the recommendation systems may suffer overfitting, and their performance may degrade accordingly, especially in the initial phase of deployment. Although real-world recommendation systems are usually updated in an online manner with the assist of increasing observed user behavior Rendle & Schmidt-Thieme (2008); Agarwal et al. (2010); He et al. (2016), introducing a feedback mechanism during their training phases can potentially improve the efficiency of the initial systems. However, this is neglected by existing learning frameworks.

Motivated by the above observations, we propose a novel framework that achieves collaborative filtering with a synthetic feedback loop (CF-SFL). As shown in Figure 1, the proposed framework consists of a "recommender" and a "virtual user." The recommender is a collaborative filtering (CF) model, that predicts items from observed user behavior. The observed user behavior reflects intrinsic preferences of users, while the recommended items represent the potential user preferences estimated by the model. Regarding the fusion of the observed user behavior and the recommended items as inputs, the virtual user, which is the key of our model, imitates real-world scenarios and synthesizes user feedback. In particular, the virtual user contains a reward estimator and a feedback generator: the reward estimator estimates rewards based on the fused inputs (the compatible

representation of the user observation and its recommended items), learned with a generative adversarial regularizer. The feedback generator provides feedback embeddings to augment the original user embeddings, conditioned on the estimated rewards as well as the fused inputs. Such a framework constructs a closed loop between the target CF model and the virtual user, synthesizing user feedback as side information to improve recommendation results.

The proposed CF-SFL framework can be interpreted as inverse reinforcement learning (IRL) approach, in which the recommender learns to recommend user items (policy) with the estimated guidance (feedback) from the proposed virtual user. The proposed feedback loops can be understood as an effective rollout procedure for recommendation, jointly updating the recommender (policy) and the virtual user (the reward estimator and the feedback generator). Eventually, even if side information (*i.e.*, real-world user feedback) is unobservable, our algorithm is still applicable to synthesize feedback in both the training and inference phases. The proposed framework

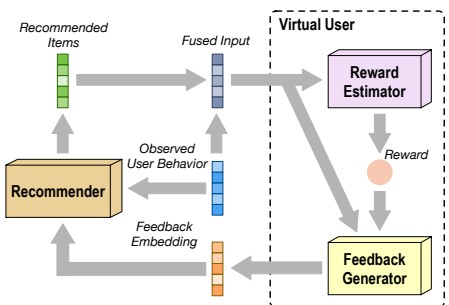

Figure 1: Illustration of our proposed CF-SFL framework for collaborative filtering.

is general and compatible with most CF methods. Experimental results show that the performance of existing approaches can be remarkably improved within the proposed framework.

## 2 PROPOSED FRAMEWORK

In this section, we first describe the problem we are interested in and give a detail description of each module that is included in the framework.

### 2.1 PROBLEM STATEMENT

Suppose we have $N$ users with $M$ items in total, we denote the observed user-item matrix as $\boldsymbol{X} = [\boldsymbol{x}_i] \in \{0,1\}^{N \times M}$, where each vector $\boldsymbol{x}_i = [x_{ij}] \in \mathcal{R}^M$, $i = 1, ..., N$, represents observed user behaviors. $x_{ij} = 1$ indicates the the $j$-th item is bought or reviewed via the $i$-th user and otherwise the $j$-th item can either be irrelevant to the $i$-th user or we have not enough knowledge about their relationship. The desired recommendation system aims to predict each user's preference, denoted as $\boldsymbol{a}_i = [a_{ij}] \in \mathcal{R}^M$, whose element $a_{ij}$ indicates the preference of the $i$-th user to the $j$-th item. Accordingly, the system recommends each user with the items having large $a_{ij}$'s.

Ideally, for each user, $\boldsymbol{x}_i$ just contains partial (actually, very sparse) information about user preference and a practical recommendation system works dynamically with a closed loop — users often generate feedback to the recommended items while the recommendation system considers these feedback to revise recommended items in the future. Therefore, we can formulate the whole recommendation process as

$$\boldsymbol{a}_i^t = \pi(\boldsymbol{x}_i, \boldsymbol{v}_i^t), \quad \boldsymbol{v}_i^{t+1} = f(\boldsymbol{x}_i, \boldsymbol{a}_i^t), \quad \text{for } i = 1, ..., N, \tag{1}$$

where $\pi(\cdot)$ represents the target recommender while $f(\cdot)$ represents the coupled feedback mechanism of the system. $\boldsymbol{v}_i \in \mathcal{R}^d$ is the embedding of user feedback to historical recommended items. At each time $t$, the recommender predicts preferred items according to observed user behaviors and previous feedback, and the user generates feedback to the recommender. Note that (1) is different from existing sequential recommendation models Mishra et al. (2015); Wang et al. (2016) because those methods ignore the feedback loop as well, which just updates recommender $\pi$ according to observed sequential observations, *i.e.*, $\boldsymbol{x}_i^t$ for different time $t$'s.[1]

Unfortunately, the feedback information is often unavailable in the training and inference phases. Accordingly, most of existing collaborative filtering-based recommendation methods ignore the feedback loop in the system, learning the target system purely from static observation user-item matrix $\boldsymbol{X}$ Liang et al. (2018); Li & She (2017). Although in some scenarios side information like user reviewers is associated with the observation matrix, the methods using such information often treat it as a kind of static knowledge rather than dynamic feedback. They mainly focus on fitting the

---

[1]When the static observation $\boldsymbol{x}_i$ in (1) is replaced with sequential observation $\boldsymbol{x}_i^t$, (1) is naturally extended to a sequential recommendation system with a feedback loop. In this work, we focus on the case with static observations and train a recommender system accordingly.

groundtruth recommended items with the recommender $\pi(\cdot)$ given fixed $\boldsymbol{x}_i$'s and $\boldsymbol{v}_i$'s, while ignore the imitation of the whole recommendation-feedback loop in (1). Without the feedback mechanism $f(\cdot)$, $\pi(\cdot)$ tends to over-fit the observed user behavior and static side information, which may degrade in practical dynamical scene.

To overcome the problems mentioned above, we propose a collaborative filtering framework with a synthetic feedback loop (CF-SFL), which explains the whole recommendation process from a viewpoint of reinforcement learning. As shown in Figure 1, besides traditional recommendation module the proposed framework further introduces a *virtual user*, which imitates the recommendation-feedback loop even if the real-world feedback are unobservable.

## 2.2 RECOMMENDER

In our framework, the recommender implements the function $\pi(\cdot)$ in (1), which takes the observed user behavior $\boldsymbol{x}_i$ and his/her previous feedback embedding $\boldsymbol{v}_i^t$ as input and recommends items accordingly. In principle, the recommender $\pi(\cdot)$ can be defined with high flexibility, which can be arbitrary collaborative filtering methods that predicting items from user representations, such as WMF Hu et al. (2008), CDAE Wu et al. (2016), VAE Liang et al. (2018) etc. In this work, we formulate the recommender from the viewpoint of reinforcement learning.

In particular, the recommendation-feedback loop generates a sequence of interactions between each user and the recommender, *i.e.*, $(\boldsymbol{s}_i^t, \boldsymbol{a}_i^t)_{t=1}^T$ for $i = 1, ..., N$. Here, $\boldsymbol{s}_i^t = [\boldsymbol{x}_i; \boldsymbol{v}_i^t]$ is the representation of user $i$ at time $t$, which is a sample in the state space $\mathcal{S}$ describing user preference. $\boldsymbol{a}_i^t$ indicates the recommended items provided by the recommender, which is a sample in the action space $\mathcal{A}$ of the recommender. Accordingly, we can model the recommendation-feedback loop as a Markov Decision Process (MDP) $\mathcal{M} = \langle \mathcal{S}, \mathcal{A}, P, R \rangle$, where $P : \mathcal{S} \times \mathcal{A} \times \mathcal{S} \mapsto \mathbb{R}$ is the transition probability of user preference and $R : \mathcal{S} \times \mathcal{A} \mapsto \mathbb{R}$ is the reward function used to evaluate recommended items. The recommender $\pi(\cdot)$ works as a policy parametrized by $\boldsymbol{\theta}$, *i.e.*, $\pi_{\boldsymbol{\theta}}(\boldsymbol{a}|\boldsymbol{s})$, which corresponds to the distribution of items conditioned on user preference. The target recommender should be an optimal policy that maximizes the expected reward: $J(\pi_{\theta}) = \sum_{t=1}^{T} \mathbb{E}_{\pi_{\boldsymbol{\theta}}} [R(\boldsymbol{s}^t, \boldsymbol{a}^t)]$, where $R(\boldsymbol{s}^t, \boldsymbol{a}^t)$ means the reward over the state-action pair $(\boldsymbol{s}^t, \boldsymbol{a}^t)$. For the $i$-th user, given $\boldsymbol{s}_i^t$, the recommender selects potentially-preferred items via

$$\boldsymbol{a}_i^t = \arg \max_{\boldsymbol{a} \in \mathcal{A}} \pi_{\theta}(\boldsymbol{a}|\boldsymbol{s}_i^t). \tag{2}$$

Note that different from traditional reinforcement learning tasks, in which both $\mathcal{S}$ and $\mathcal{A}$ are available while $P$ and $R$ are with limited accessibility, our recommender just receives partial information of state — it does not observe users' feedback embedding $\boldsymbol{v}_i$. In other words, to optimize the recommender, we need to build a reward model and a feedback generator jointly, which motivates us to introduce a virtual user into the framework.

## 2.3 VIRTUAL USER

The virtual user module aims to implement the feedback function $f(\cdot)$ in (1), which not only models the reward of the items provided by the recommender but also generates feedback to complete the representations of state. Accordingly, the virtual user contains the following two modules:

**Reward Estimator** The reward estimator parametrizes the function of reward, which takes the current prediction $\boldsymbol{a}_i^t$ and user preference $\boldsymbol{s}_i^t$ as input and evaluate their compatibility. In this work, we implement the estimator with parameter $\boldsymbol{\phi}$, which is defined as

$$R_{\boldsymbol{\phi}}(\boldsymbol{s}_i^t, \boldsymbol{a}_i^t) = \text{sigmoid}(g(h(\boldsymbol{x}_i, \boldsymbol{a}_i^t))). \tag{3}$$

In this work, we use the static part of the state $\boldsymbol{s}_i^t$, *i.e.*, the observed user behaviors $\boldsymbol{x}_i$ as input. $h(\cdot, \cdot)$ is the fusion function which merges $\boldsymbol{x}_i$ and $\boldsymbol{a}_i^t$ into a real value vector (the fused input is shown in Figure 5 and described in Appendix). $g(\cdot)$ is the single value regression function that translates the fused input into a single reward value. The sigmoid function is used to restrict the regression value between 0 and 1.

**Feedback Generator** The feedback generator connects the reward estimator with the recommender via generating a feedback embedding, *i.e.*,

$$\boldsymbol{v}_{t+1}^i = F_{\boldsymbol{\psi}}(h(\boldsymbol{x}_i, \boldsymbol{a}_i^t), R(\boldsymbol{s}_i^t, \boldsymbol{a}_i^t)), \tag{4}$$

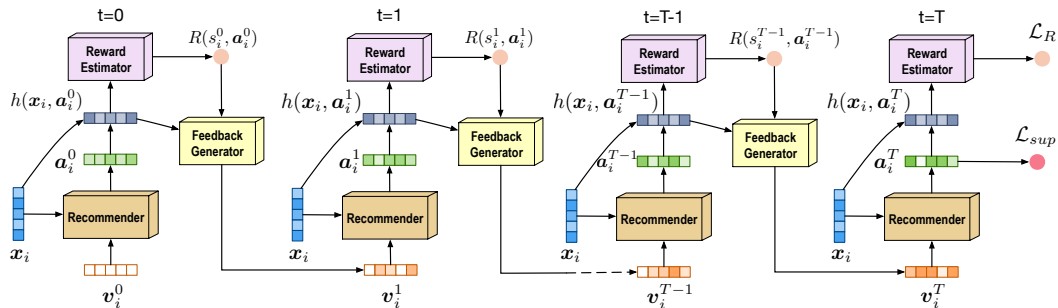

Figure 2: Unrolling the recurrent CF-SFL framework into an iterative learning process with multiple time steps T.

where $\psi$ represents the parameters of the generator. Specifically, the a parametric function $F_\psi(\cdot, \cdot)$ considers the fused input and the estimated reward and returns a feedback embedding $\boldsymbol{v}_t^i \in R^d$ to the recommender. $R(\boldsymbol{s}_i^t, \boldsymbol{a}_i^t)$ indicates the compatibility between the recommended items and user preferences, and $h(\boldsymbol{x}_i, \boldsymbol{a}_i^t)$, which is a vector rather than a scalar like reward, further enriches the information of the reward to generate feedback embeddings. Consequently, the recommender receives the informative feedback as a complementary component of the static observation $\boldsymbol{x}_i$ to make an improved recommendation via (2).

## 3 LEARNING ALGORITHM

### 3.1 LEARNING TASK

Based on the proposed framework, we need to jointly learn the policy corresponding to the recommender $\pi_\theta$, the reward estimator $R_\phi$ and the feedback generator $F_\psi$. Suppose that we have a set of labeled samples $\mathcal{D} = \{\boldsymbol{x}_i, \tilde{\boldsymbol{a}}_i\}$, where $\boldsymbol{x}_i \in \mathbb{R}^M$ is the historical behaviors of user $i$ derived from the user-item matrix $\boldsymbol{X}$ and $\tilde{\boldsymbol{a}}_i$ is the ground truth of the recommended item for the user based on his/her behavior $\boldsymbol{x}_i$. We formulate the learning task as the following min-max optimization problem.

$$\min_{\pi_\theta, F_\psi} \max_{R_\phi} \mathcal{L}(\pi_\theta, R_\phi, F_\psi) \tag{5}$$

where

$$\mathcal{L}(\pi_\theta, R_\phi, F_\psi) = \underbrace{\sum_i \mathcal{L}_{\sup}(\boldsymbol{a}_i, \tilde{\boldsymbol{a}}_i; \pi_\theta, F_\psi)}_{\text{supervised loss}} - \underbrace{\mathbb{E}_{\boldsymbol{a} \sim \pi_\theta}[\log(R_\phi(\boldsymbol{s}, \boldsymbol{a}))] - \mathbb{E}_{\tilde{\boldsymbol{a}} \sim \mathcal{D}}[1 - \log(R_\phi(\boldsymbol{s}, \tilde{\boldsymbol{a}}))]}_{\text{Collaboration with adversarial regularizer}}. \tag{6}$$

In particular, the first term $\mathcal{L}_{\sup}$ in (6) can be any supervised loss based on labeled data $\mathcal{D}$, *e.g.*, the evidence lower bound (ELBO) proposed in VAEs Liang et al. (2018) (and used in our work). This term ensures the recommender to fit the groundtruth labeled data. The second term considers the following two types of interactions among these three modules:

- The *collaboration* between the recommender policy $\pi_\theta$ and the feedback generator $F_\psi$ towards a better predictive recommender.
- The *adversarial game* between the recommender policy $\pi_\theta$ and the reward estimator $R_\phi$.

Accordingly, given current reward model, we update the recommender policy $\pi_\theta$ and the feedback generator $F_\psi$ to maximize the expected reward derived from the generated user preference $\boldsymbol{s}$ and the recommended item $\boldsymbol{a}$. On the contrary, given the recommended policy and the feedback generator, we improve the reward estimator $R_\phi$ by sharpening its criterion — the updated reward estimator maximizes the expected reward derived from the generated user preference and the ground truth of item while minimize the expected reward based on the recommended item. Therefore, we solve (5) via alternating optimization. The updating of $\pi_\theta$ and $F_\psi$ is achieved by minimizing

$$\mathcal{L}_C(\pi_\theta, F_\psi) = \sum_i \mathcal{L}_{\sup}(\boldsymbol{a}_i, \tilde{\boldsymbol{a}}_i; \pi_\theta, F_\psi) - \mathbb{E}_{\boldsymbol{a} \sim \pi_\theta}[\log(R_\phi(\boldsymbol{s}, \boldsymbol{a}))]. \tag{7}$$

And the updating of $R_\phi$ is achieved by maximizing

$$\mathcal{L}_A(R_\phi) = -\mathbb{E}_{\boldsymbol{a} \sim \pi_\theta}[\log(R_\phi(\boldsymbol{s}, \boldsymbol{a}))] - \mathbb{E}_{\tilde{\boldsymbol{a}} \sim \mathcal{D}}[1 - \log(R_\phi(\boldsymbol{s}, \tilde{\boldsymbol{a}}))] \tag{8}$$

Both these two updating steps can be solved effectively via stochastic gradient descent.

### 3.2 Unrolling for learning and inference

Because the proposed framework contains a closed loop among learnable modules, during training we unroll the loop and let the recommender interact with the virtual user in $T$ steps. Specifically, at the initial stage, the recommender takes the observed user behaviour $\boldsymbol{x}_i$ and an all-zero initial feedback embedding $\boldsymbol{v}_i^0$, to make recommendations. At each step $t$, the recommender predicts the items $\boldsymbol{a}_i^t$ given $\boldsymbol{x}_i$ and $\boldsymbol{v}_i^t$ to the virtual user, and receives the feedback embedding $\boldsymbol{v}_i^{t+1}$. The loss is defined according to the output of the last step, $i.e.$, $\boldsymbol{a}^T$ and $\boldsymbol{v}^T$, and the modules are updated accordingly. After the model is learned, in the testing phase we need to infer the recommended item in the same manner, unrolling the feedback loop and deriving $\boldsymbol{a}^T$ as the final recommended item. The details of unrolling process are illustrated in Figure 2, and the detailed scheme of our learning algorithm is shown in Algorithm 1 in appendix.

## 4 CF-SFL as Inverse Reinforcement Learning

Our CF-SFL framework automatically discovers informative user feedback as side information and gradually improve the training for the recommender. Theoretically, it closely connects with Inverse Reinforcement Learning (IRL). Specifically, we jointly learn the reward function $R(\cdot, \cdot)$ and the policy (recommender) $\pi(\cdot, \cdot)$ from the $expert\ trajectories$ $\mathcal{D}_E$ (the observed labeled data). $\mathcal{D}_E$ typically consists of state-action pairs generated from an expert policy $\pi_E$ with the corresponding environment dynamics $\rho_E$. And the goal of the IRL is to recover the optimal reward function $R^*(\cdot, \cdot)$ as well as the corresponding recommender $\pi^*$. Formally, the IRL is defined as:

$$\{R^*, \pi^*\} \triangleq \text{IRL}(\pi_E) = \arg \max_{R \in \mathbb{R}^{\mathcal{S} \times \mathcal{A}}} \sum_{\boldsymbol{s}, \boldsymbol{a}} \rho_E(\boldsymbol{s}, \boldsymbol{a}) R(\boldsymbol{s}, \boldsymbol{a}) - [\max_{\pi \in \Pi} H(\pi) + \sum_{\boldsymbol{s}, \boldsymbol{a}} \rho(\boldsymbol{s}, \boldsymbol{a}) R(\boldsymbol{s}, \boldsymbol{a})] \quad (9)$$

$$= \arg \max_{r \in \mathbb{R}^{\mathcal{S} \times \mathcal{A}}} \min_{\pi \in \Pi} \underbrace{-H(\pi) + \sum_{\boldsymbol{s}, \boldsymbol{a}} \rho_E(\boldsymbol{s}, \boldsymbol{a}) R(\boldsymbol{s}, \boldsymbol{a}) - \sum_{\boldsymbol{s}, \boldsymbol{a}} \rho(\boldsymbol{s}, \boldsymbol{a}) R(\boldsymbol{s}, \boldsymbol{a})}_{\mathcal{L}(\pi, R)} \quad (10)$$

Intuitively, the objective enforces the expert policy $\pi_E$ to induce higher rewards (the $\max$ part), than all other policies. This objective is sub-optimal if the expert trajectories are noisy, $i.e.$, the expert is not perfect and its trajectories are not optimal. This will render the learned policy always performs worse than the expert one. Besides, the illed-defined IRL objective often induces multiple solutions due to flexible solution space, $i.e.$, one can assign an arbitrary reward to trajectories not from expert, as long as these trajectories yields lower rewards than the expert trajectories. To alleviate these issues, some constraints are placed into the objective functions, $e.g.$, a convex reward functional, $\psi : \mathbb{R}^{\mathcal{S} \times \mathcal{A}} \rightarrow \mathbb{R}$, which usually works as a regularizer.

$$\{R^*, \pi^*\} = \arg \max_{R \in \mathbb{R}^{\mathcal{S} \times \mathcal{A}}} \min_{\pi \in \Pi} \mathcal{L}(\pi, R) - \psi(R). \quad (11)$$

To imitate the expert policy and provide better generalization, we adopt the adversarial regularizer Ho & Ermon (2016), which defines $\psi$ with the following form:

$$\psi(R) \triangleq \begin{cases} \mathbb{E}_{\pi_E}\left[q(R(\boldsymbol{s}, \boldsymbol{a}))\right] & \text{if } R(\boldsymbol{s}, \boldsymbol{a}) \geq 0 \\ +\infty & \text{otherwise} \end{cases},$$

where $q(x) = x - \log(1 - e^{-x})$. This regularizer places low penalty on reward functions $R$ that assign an amount of positive value to expert state-action pairs; however, if $R$ assigns low value (close to zero, which is the lower bound) to the expert, then the regularizer will heavily penalize $R$. With induced adversarial regularizer, we obtain a new imitation learning algorithm for recommender:

$$\min_{\theta} \psi^*(\rho_\pi - \rho_{\pi_E}) - \lambda H(\pi_\theta) \quad (12)$$

Intuitively, we want to find a saddle point $(R_\phi, \pi_\theta)$ of the expression:

$$\mathbb{E}_{\pi_\theta}[\log(R(\boldsymbol{s}, \boldsymbol{a}))] + \mathbb{E}_{\pi_E}[1 - \log(R(\boldsymbol{s}, \boldsymbol{a}))] - \lambda H(\pi_\theta), \quad (13)$$

where $R(\boldsymbol{s}, \boldsymbol{a}) \in (0, 1)$. Note equation 11 is derived from the objective of traditional IRL. However, distinct from the traditional approach, we propose a feedback generator to provide feedbacks to the recommender. In terms of the reward estimator, it tends to assign lower rewards to the predicted results by the recommender $\pi_\theta$ and higher rewards for the expert policy $\pi_E$, which aims to discriminate $\pi_\theta$ from $\pi_E$:

$$\mathcal{L}_R = \mathbb{E}_{\pi_\theta}[\log(R(\boldsymbol{s}, \boldsymbol{a}))] + \mathbb{E}_{\pi_E}[1 - \log(R(\boldsymbol{s}, \boldsymbol{a}))] \quad (14)$$

Similar to standard IRL, we update the generator to maximize the expected reward with respect to $\log R(\boldsymbol{s}, \boldsymbol{a})$, moving towards expert-like regions of user-item space. In practice, we incorporate feedback embedding to update the user preferences, and the objective of the recommender is:

$$\mathcal{L}_F = \mathbb{E}_{\pi_\theta}[-\log(R([\boldsymbol{x}_i, \boldsymbol{v}_i^t], \boldsymbol{a}))] - \lambda H(\pi_\theta) \tag{15}$$

where $\boldsymbol{v}_i^t = F_\psi(h(\boldsymbol{x}_i, \boldsymbol{a}_i^t), R(\boldsymbol{s}_i^t, \boldsymbol{a}_i^t))$.

## 5 RELATED WORK

**Collaborative Filtering.** Collaborative filtering (CF) can be roughly categorized into two groups: CF with implicit feedback Bayer et al. (2017); Hu et al. (2008) and those with explicit feedback Koren (2008); Liu et al. (2010). In implicit CF, user-item interactions are binary in natural (*i.e.*, 1 if clicked and 0 otherwise) as oppose to explicit CF where item ratings (e.g., 1-5 stars) are typically the subject of interests. Implicit CF has been widely studied, examples including factorization of user-item interactions He et al. (2016); Koren (2008); Liu et al. (2016); Rendle (2010); Rennie & Srebro (2005) and ranking based approach Rendle et al. (2009). And our CF-SFL is a new framework for implicit CF.

Currently, neural network based models have achieved state-of-the-art performance for various recommender systems Cheng et al. (2016); He et al. (2018; 2017); Zhang et al. (2018); Liang et al. (2018). Among these methods, NCF He et al. (2017) casts the well-established matrix factorization algorithm into an entire neural framework, combing the shallow inner-product based learner with a series of stacked nonlinear transformations. This method outperforms various of traditional baselines and has motivated many following works such as NFM He et al. (2017), Deep FM Guo et al. (2017) and Wide and Deep Cheng et al. (2016). Recently, deep generative has achieved remarkable success. VAEs Liang et al. (2018) uses variational inference to scale up the algorithm for large-scale dataset and has shown significant success in recommender systems with a usage of multinormial likelihood. Our CF-SFL is a general framework that can adapt to these models seamlessly.

**RL in Collaborative Filtering.** For RL-based methods, contextual multi-armed bandits are firstly utilized to model the interactive nature of recommender systems. Thompson Sampling (TS) Chapelle & Li (2011); Kveton et al. (2015); Zhang et al. (2017) and Upper Confident Bound (UCB) Li et al. (2010) are used to balance the trade-off between exploration and exploitation. Zhao et al. (2013) combine matrix factorization with bandits to include latent vectors of items and users for better exploration. The MDP-Based CF model can be viewed as a partial observable MDP (POMDP) with partial observation of user preferences Sunehag et al. (2015). Value function approximation and policy based optimization can be employed to solve the MDP. Zheng et al. (2018) and Taghipour & Kardan (2008) modeled web page recommendation as a Q-Learning problem and learned to make recommendations from web usage data. Sunehag et al. (2015) introduced agents that successfully address sequential decision problems. Zhao et al. (2018) propose a novel page-wise recommendation framework based on deep reinforcement learning. In this paper, we consider the recommending procedure as sequential interactions between virtual users and recommender; and leverage feedbacks from virtual users to improve the recommendation. Recently, Chen et al. (2019) proposed an off-policy corrections technique, and successfully applied it in real-world applications.

## 6 EXPERIMENTS

**Datasets** We investigate the effectiveness of the proposed CF-SFL framework on three benchmark datasets of recommendation systems. (*i*) MovieLens-20M (ML-20M), a movie recommendation service contains tens of millions user-movie ratings. (*ii*) Netflix-Prize (Netflix), another user-movie ratings dataset collected

Table 1: Basic information of the considered datasets.

|  | **ML-20M** | **Netflix** | **MSD** |
|---|---|---|---|
| # of users | 136,677 | 463,435 | 571,355 |
| # of items | 20,108 | 17,769 | 41,140 |
| # of interactions | 10.0M | 56.9M | 33.6M |
| # of held-out-users | 10.0K | 40.0K | 50.0K |
| % of sparsity | 0.36% | 0.69% | 0.14% |

by the Netflix Prize Bennett & Lanning (2007). (*iii*) Million Song Dataset (MSD), a user-song rating dataset, which is released as part of the Million Song Dataset Bertin-Mahieux et al. (2011). To directly compare with existing work, we employed the same pre-processing procedure as Liang et al. (2018). A summary statistics of these datasets are provided in Table 1.

Table 3: Performance comparison between our CF-SFL framework and various of baselines. VAE* is the results based on our own runs and VAE† is the VAE model with our reward estimator.

| Methods | ML-20M | | | Netflix | | | MSD | | |
|---|---|---|---|---|---|---|---|---|---|
| | R@20 | R@50 | NDCG@100 | R@20 | R@50 | NDCG@100 | R@20 | R@50 | NDCG@100 |
| SLIM | 0.370 | 0.495 | 0.401 | 0.347 | 0.428 | 0.379 | - | - | - |
| WMF | 0.360 | 0.498 | 0.386 | 0.316 | 0.404 | 0.351 | 0.211 | 0.312 | 0.257 |
| CDAE | 0.391 | 0.523 | 0.418 | 0.343 | 0.428 | 0.376 | 0.188 | 0.283 | 0.237 |
| aWAE | 0.391 | 0.532 | 0.424 | 0.354 | 0.441 | 0.381 | - | - | - |
| VAE | 0.395 | 0.537 | 0.426 | 0.351 | 0.444 | 0.386 | 0.266 | 0.364 | 0.316 |
| VAE* | 0.395 | 0.535 | 0.425 | 0.350 | 0.444 | 0.386 | 0.260 | 0.356 | 0.311 |
| VAE† | 0.396 | 0.536 | 0.426 | 0.352 | 0.445 | 0.387 | 0.263 | 0.360 | 0.314 |
| CF-SFL | **0.404** | **0.542** | **0.435** | **0.355** | **0.449** | **0.394** | **0.273** | **0.369** | **0.323** |

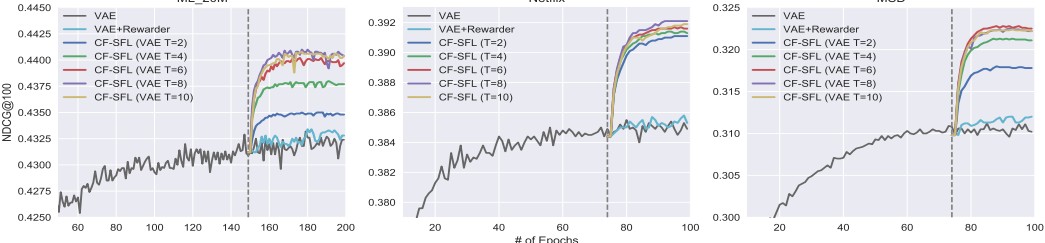

Figure 3: Performance (NDCG@100) boosting on the validation sets.

**Evaluation Metrics**   We employ Recall@r[2] together with NDCG@r[3] as the evaluation metric for recommendation, which measures the similarity between the recommended items and the ground truth. Recall@r considers top-r recommended items equally, while NDCG@r ranks the top-r items and emphasizes the importance of the items that are with high ranks.

**Set-up**   For our CF-SFL framework, the architectures of its recommender, reward estimator and feedback generator are shown in Table 2. To represent the user preference, we normalize $x_i$ and $v_i^t (t > 0)$ independently and concatenate the two into a single vector. To learn the model, we pre-train the recommender (150

Table 2: Architecture of our CF-SFL framework.

| Recommender | Reward Estimator | Feedback Generator |
|---|---|---|
| Input $\mathcal{R}^M$ | Input $\mathcal{R}^{64}$ | Input $\mathcal{R}^{65}$ |
| $M \times 600$, tanh | $64 \times 128$, ReLU | $64 \times 128$, ReLU |
| $600 \times 200$ (x2) | $128 \times 128$, ReLU | |
| Sample $\mathcal{R}^{200}$ | | $128 \times 128$, ReLU |
| $200 \times 600$, tanh | $128 \times 128$, ReLU | |
| $600 \times M$ softmax | $128 \times 1$, sigmoid | $128 \times 128$ |

epochs for ML-20M and 75 epochs for Netflix and MSD) and optimize the entire framework (50 epochs for ML-20M and 25 epochs for the other two). $\ell_2$ regularization with a penalty term 0.01 is applied to the recommender, and Adam optimizer Kingma & Ba (2014) with batch in size of 500 is employed. The framework is built with Tensorflow and will be publicly available upon publication.

**Baselines**   To demonstrate the superiority of our framework, we consider multiple state-of-the-art approaches as baselines, which can be categorized into two types: (i) Linear based models: SLIM Ning & Karypis (2011) and WMF Hu et al. (2008). (ii) Deep neural network based models: CDAE Wu et al. (2016), VAE Liang et al. (2018) and aWAE Zhong & Zhang (2018). It should be noted that our CF-SFL is a generalized framework, which is compatible with all these approaches. In particular, as shown in Table 2, we implement our recommender as the VAE-based model Liang et al. (2018) for a fair comparison. In the following experiments, we will show that besides such a setting the recommender can be implemented by other existing models as well.

**Performance Analysis**   All the evaluation metrics are averaged across all the test sets.
*(i)* Quantitative results: we test various methods and report their results in Table 3. With the proposed CF-SFL framework, we significantly improve the performance of the baselines on all the evaluation metrics. These experimental results demonstrate the power of the proposed CF-SFL framework, which provides informative feedback as the side information. Particularly, we observed that the performance between the base model (VAE*) is similar to that of its variation with the reward estimator (VAE†). It implies that simply learning a feedback from the reward estimator via back-propagation

---
[2]https://en.wikipedia.org/wiki/Precision_and_recall
[3]https://en.wikipedia.org/wiki/Discounted_cumulative_gain

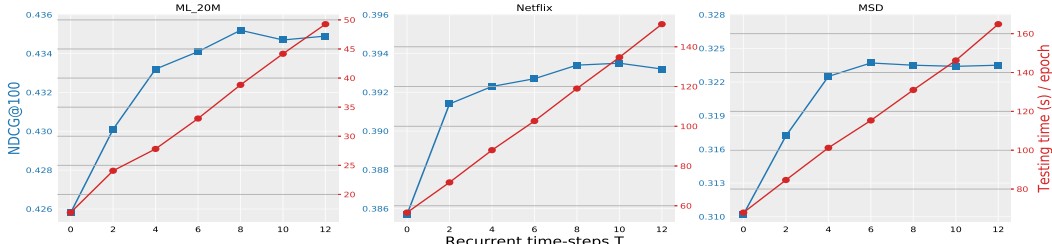

Figure 4: The blue curve summarizes NDCG@100 and red curves report the computational cost for model inference in each epoch. In each sub-figure, we vary the time steps from 0 to 12 (T=0 is the base recommender).

is inefficient. Compared with such a naive strategy, the proposed CF-SFL provides more informative feedback to the recommender, and thus, is able to improve recommendation results more effectively.
*(ii)* Learning Comparison: In Figure 3, we show the training trajectory of the baselines (VAE, VAE+reward estimator) and the CF-SFL with multiple time steps. There are several interesting findings. (a) The performance of the base VAE doesn't improve after the pre-training steps, *e.g.,* 75 epochs for Netflix. In comparison, the proposed CF-SFL framework can further improve the performance once the whole model is triggered. (b) The CF-SFL yields fast convergence once the whole framework is activated. (c) Coincide with results in Table 3, the trajectory of VAE$^\dagger$ in Figure 3, is similar to that of the base VAEs (VAE$^*$). In contrast, the trajectories of our CF-SFL methods are more smooth and able to converge to a better local minimum. This phenomenon further verifies that our CF-SFL learns informative user feedback with better stability. (d) With an increasing of time steps $T$ in a particular range ($T \leq 8$ for ML-20M), CF-SFL achieves faster and better performance. One possible explanation is the learning with our unrolled structure — parameters are shared across different time-steps, and a more accurate gradient is found towards the local minimum. (e) We find ML-20M and MSD are more sensitive to the choice of $T$ when compared with Netflix. Therefore, the choice of $T$ should adjust to different datasets.
*(iii)* CF-SFL with dynamic time steps: As shown in Figure 2, the learning of CF-SFL involves a recurrent structure with $T$ times steps. We investigate the choice of $T$ and report its influence on the performance of our method. Specifically, the NDCG@100 with different $T$'s is shown in Figure 4. Within 6 time steps, CF-SFL consistently boots the performance on all the three datasets. Even with a larger time steps, the results remain stable. Additionally, the inference time of CF-SFL is linear on time steps $T$. To achieve a trade-off between performance and efficiency, in our experiments we set $T$ to 8 for ML-20M and Netflix and 6 for MSD.

**Generalization Study** As aforementioned, our CF-SFL is a generalized framework which is compatible with many existing collaborative filtering approaches. We study the usefulness of our CF-SFL on different recommenders and present the results in Table 4. Specifically, two types of recommenders are being considered: the linear approaches like WARP Weston et al. (2011) and MF Hu et al.

Table 4: Performance of our CF-SFL with various of recommenders are reported.

| Recommender | w/o CF-SFL | w CF-SFL | Gain ($10^{-3}$) |
|---|---|---|---|
| WARP | 0.31228 | 0.33987 | +27.59 |
| MF | 0.41587 | 0.41902 | +3.15 |
| DAE | 0.42056 | 0.42307 | +2.51 |
| VAE | 0.42546 | 0.43472 | +9.26 |
| VAE-(Gaussian) | 0.42019 | 0.42751 | +7.32 |
| VAE-($\beta = 0$) | 0.42027 | 0.42539 | +5.02 |
| VAE-Linear | 0.41563 | 0.41597 | +0.34 |

(2008), and deep learning methods, *e.g.*, DAE Liang et al. (2018) and the variation of VAE in Liang et al. (2018). We find that our CF-SFL is capable of generalizing to most of the exisiting collaborative filtering approaches and boosts their performance accordingly. The gains achieved by our CF-SFL may vary depending on the choice of recommender.

## 7 CONCLUSION

We propose a CF-SLF framework to simulate user feedback. It constructs a virtual user to provide informative side information as user feedback. Mathematically we formulate the framework as an IRL problem and learn the optimal policy by feeding back the action and reward. Specially, a recurrent architecture was built to unrolled the framework for efficient learning. Empirically we improve the performance of state-of-the-art collaborative filtering method with a non-trivial margin. Our framework serves as a practical solution making IRL feasible over large-scale collaborative filtering. And it will be interesting to investigate the framework in other applications, such as sequential recomender systems *etc*.

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

## A APPENDIX

---

**Algorithm 1:** CF-SFL training with stochastic optimization

---

1: **Input:** A user-item matrix $X$ and labeled pairs $\mathcal{D}$, the unrolling step $T$, the size of batch $b$.
2: **Output:** Recommender $\pi_{\theta}$, reward estimator $R_{\phi}$, and feedback generator $F_{\psi}$
3: **Initialization**: randomly initialize $\theta$, $\phi$ and $\psi$;
   /* stage 1: pretrain the recommender */
4: **while** not converge **do**
5:    Sample a batch of $\{x_i, \tilde{a}_i\}_{i=1}^{b}$ from $\mathcal{D}$;
6:    Update $\theta$ via minimizing $\mathcal{L}_{sup}$.
7: **end while**

   /* stage 2: pretrain the reward estimator */
8: **while** not converge **do**
9:    Sample a batch of $\{x_i, \tilde{a}_i\}_{i=1}^{b}$ from $\mathcal{D}$ and calculate $\{R(s_i, \tilde{a}_i)\}_{i=1}^{b}$;
10:   Sample another batch of user $\{x_i\}_{i=1}^{b}$ and set $v_i = \mathbf{0}$
11:   Infer the recommended items $\{a_i\}_{i=1}^{b}$ and calculate $\{R(s_i, a_i)\}_{i=1}^{b}$;
12:   Update $\phi$ via maximizing (8).
13: **end while**

   /* stage 3: alternative train all the modules */
14: **while** not converge **do**
15:   Sample a batch of $\{x_i, \tilde{a}_i\}_{i=1}^{b}$ from $\mathcal{D}$, initialize feedback embedding $v^0 = \mathbf{0}$;

      /* Update recommender and feedback generator */
16:   Feed $\{x_i\}_{i=1}^{b}$ and $V^{(0)}$ into the recommender and infer $\{a_i^T\}_{i=1}^{b}$ through a $T$-step recurrent structure.
17:   Collect the corresponding reward $\{R(s_i^T, a_i^T)\}_{i=1}^{b}$
18:   Update $\theta$ and $\psi$ via minimizing (7).

      /* Reward estimator update step */
19:   Sample a batch of $\{x_i, \tilde{a}_i\}_{i=1}^{b}$ from $\mathcal{D}$, and calculate $\{R(s_i, \tilde{a}_i)\}_{i=1}^{b}$;
20:   Sample a batch of $\{x_i\}_{i=1}^{b}$, infer the recommended items $\{a_i^T\}_{i=1}^{b}$ and calculate $\{R(s_i, a_i^T)\}_{i=1}^{b}$;
21:   Update $\phi$ via maximizing (8)
22: **end while**

---

### A.1 FUSION FUNCTION

Here we give a detail description of the fused function we have proposed. A straight-forward way to build the fusion function $h(x_i, a_i)$ is to concatenate $x_i$ and $a_i$, and feed it into a linear layer to learn a lower dimensional representation. However, in practice this method is infeasible since the dimension of items, $M$, is extremely high and the usage of the concatenation will make the problem even worse. To this end, we introduce a sparse layer. This layer includes a lookup table $B \in \mathcal{R}^{M \times d}$. Once we have inferred the the recommended items $a_i$ based on the observation $x_i$, we build the the fused input as

$$h(x_i, a_i) = \frac{1}{|x_i|} \sum_{j=1}^{M} \delta(x_{ij}) B_j + \sum_{k=1}^{M} a_{ik} B_k \qquad (16)$$

where $\delta$ is the Dirac Delta function and takes value 1 if $x_{ij} = 1$, $|x_i|$ is number of 1 in $x_i$. The parameters of the lookup table will be automatically learned during the training phrase. We show an example to illustrate the working scheme for the proposed fusion function in Figure 5. The benefits for the proposed approach can be summarized as two folds: 1) it reduce the computational cost of the standard linear transformation under the general sparse set up and saves number of parameters in our proposed adversarial learning framework; 2) This lookup table is shared across the observation and the recommended items, building a unified space for the users existing preference and missing preference. Empirically such shared knowledge boosts the performance of our CF-SFL framework.

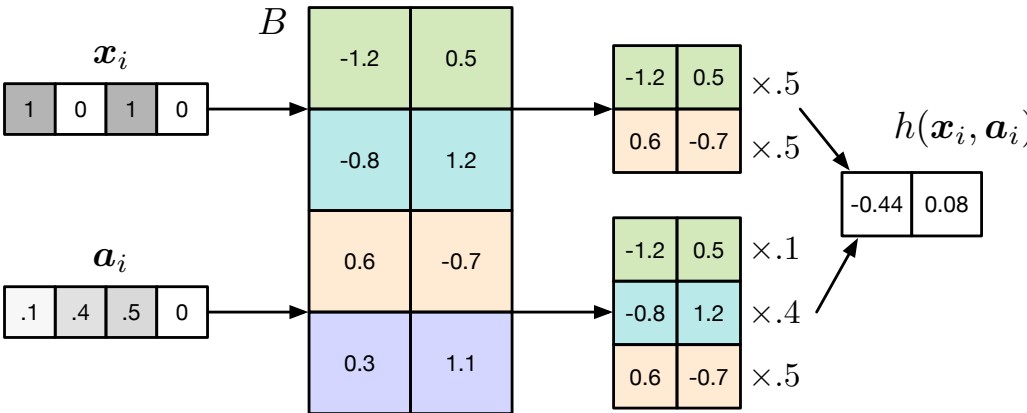

Figure 5: An example of the our fused function working scheme. The user behavior $\boldsymbol{x}_i$ and the recommended items $\boldsymbol{a}_i$ share the same lookup table $B$. $[-0.04, 0.08]$ is the fused input for the given example. This method works efficient if $\boldsymbol{x}_i$ and $\boldsymbol{a}_i$ are sparse.

