# OpenReview forum: "Collaborative Filtering With A Synthetic Feedback Loop"
_ICLR.cc/2020/Conference — Reject_

### Official Review · AnonReviewer3 · 2019-10-06
**Official Blind Review #3**

**Rating:** 6

**Review:**

The paper is essentially an attempt to incorporate a form of reinforcement learning into recommender systems, via the use of a synthetic feedback loop and a "virtual user".

Overall this seems like a nice attempt to combine Inverse Reinforcement Learning frameworks with collaborative filtering algorithms.

Without a background in reinforcement learning, it was difficult for me to assess exactly how this compares to similar reinforcement learning work, or exactly how "obvious" the technical contribution is.

Nevertheless combining reinforcement learning with recommender systems is a topic of growing interest, and it is nice to see a paper in this space that makes a contribution with strong quantitative experiments, i.e., it is able to compete with (reasonably) strong baselines.

The experiments mostly look good, with a few representative, and quite large, datasets. The baselines are perhaps not state-of-the-art but represent reasonable points of comparison and are enough for a proof-of-concept. The paper is quite thorough in terms of experimental comparisons.

Overall this seems like an interesting approach and a reasonably timely paper, making what seems like a compelling contribution to a current hot topic, and passes the bar in terms of experiments. Regarding the merits of the technical contribution, I'll perhaps have to defer to other reviewers, but overall the contribution seems above the bar.

**Experience Assessment:**

I have published one or two papers in this area.

**Review Assessment: Checking Correctness Of Derivations And Theory:**

I did not assess the derivations or theory.

**Review Assessment: Checking Correctness Of Experiments:**

I assessed the sensibility of the experiments.

**Review Assessment: Thoroughness In Paper Reading:**

I made a quick assessment of this paper.

---

### Official Review · AnonReviewer1 · 2019-10-22
**Official Blind Review #1**

**Rating:** 3

**Review:**

The paper proposes to learn a "virtual user" while learning a "recommender" model, to improve the performance of the recommender system. The "virtual user" is used to produce the "reward" signal for training the recommendation system (which is trained using RL). Jointly learning the recommender and the "virtual user" provides a synthetic feedback loop that helps to improve the performance of the recommendation system. The paper formulates the training of the "virtual user" as an inverse RL problem and uses adversarial regularizer.

The paper proposes an interesting idea but more experiments (and explanation) is needed to bring out the usefulness of the work. In general, the writing needs to be polished as well.

===============

Following are the items that need to be improved:

## Significance of the results

* The results in Tables 3 and 4 provide only marginal improvements over the baseline. These improvements do not appear to be statistically significant. It would help if the authors comment on why these results appear significant and also provide variance values/curves for the results.

## Role of the feedback general F

* Is there any separate loss for training F or is it always trained along with \pi?

* Is the feedback capturing some sort of "memory" or "past preferences/behaviors" of the user? If yes, wouldn't using a recurrent recommender also capture these effects without needing the feedback model F? Note that I am not criticizing the choice of F. I am trying to understand the role played by F (in addition to the recommender pi).

* If there is no separate loss for F, I wonder how would the performance change if the F network was to be removed and the reward value was to be fed into a recurrent recommended. (The paper seems to have considered a special case where a non-recurrent recommender is used with the reward value). The reason I am stressing on this is that one of the key distinctions of the authors' work is the use of feedback generator and it would be useful to quantify the benefits on this modification.


## Others

* How is the static representation, x, computed? From eq 16 (appendix), it appears that x is a binary vector that captures what items have been purchased/reviewed. Is that correct? If yes, wouldn't x have an enormous dimensionality?

* The recommendation system is operating in a sequential decision-making setup. The formulation of R and F do no consider any sequential information. For example, let us say that the recommender recommends the items a1, a2 and a3 in 3 timesteps. The reward at time 3 is a function of x and a3 and the information of a1 and a2 is not used.

* The loss function has many components and I want to make sure I understand what gradients flow where. So please correct me if I missed something;

    * supervised learning loss (from the real data) trains pi and F (equation 6).
    * pi and F collaborate to get a high reward from R (since we do not have the ground truth corresponding to R). No gradient flows to R.
    * Adversarial game between pi and R - which is used to update R.

* I understand that the loss is defined according to the output of the last step but do the gradients flow through all the intermediate steps?

* The training/inference procedure seems to be doing something strange. Let us assume that T = 5. So the 5 items are recommended to the "virtual user" and only the 5th item is recommended to the real user. Now the recommendation of the 5th item depends on the first 4 items that the real user has not seen.

* In equation 15, what does the subscript F stand for?

* What algorithm is used to train the policy pi?

=================


The following are the items that should be corrected in the future iterations of the paper. Please note that these things did not impact the score.

* It seems that the irrelevance of an item (for a user) is treated the same way as missing information about the relevance of the item. Could the authors discuss this more in the paper?

* Is there any reason why this approach is only used with CF methods?

* Some writing choices seem to make the problem statement more complex than it is. For example, the authors discuss how their work is different than traditional RL instead of simply stating that their work is set up as a POMDP.

* Articles are missing (or mixed) up at many places.


**Experience Assessment:**

I have read many papers in this area.

**Review Assessment: Checking Correctness Of Derivations And Theory:**

I assessed the sensibility of the derivations and theory.

**Review Assessment: Checking Correctness Of Experiments:**

I carefully checked the experiments.

**Review Assessment: Thoroughness In Paper Reading:**

I read the paper thoroughly.

---

### Official Review · AnonReviewer2 · 2019-10-24
**Official Blind Review #2**

**Rating:** 3

**Review:**

Although I assume somebody well-versed in the recent collaborative filtering literature would not have trouble, I had too much difficulty understanding the setup and the model to be able to recommend the paper for acceptance.  Perhaps if you clarify the following questions in your rebuttal and the final version of the paper, I can view the paper more favorably:

- In the Problem Statement section, it says that we assume that there is an unknown user preference vector a_i for each user with elements in reals.  How should we think about what a_i is?  Is it a binary vector? A probability distribution over items?  An arbitrary vector of reals? Nonnegative reals?
- In the problem statement, we have that "recommender predicts preferred items . . . and the user generates feedback to the recommender."  Can you make this concrete?  Would feedback be like a binary label for whether each of the recommended items was of interest?  Or whether any were of interest?
- In general, I had a very difficult time figuring out the difference between the information in x_i and "feedback".  Is it that x_i only has information about what user i selected, but no information about what the user was presented and did not select?  Is the feedback then the combination of what was presented and whether or not it was selected?
- In Section 2.2, we discuss a_i^t as the recommended items provided by the recommender, sampled from the action space A -- this seems like a different a_i from the one in the Problem Statement section?  If so, this is an unfortunate notation clash.
- If we're producing sets of items at a time, as your wording frequently indicates, should we think of the action space A as a set of sets of items?
- When describing the "recommendation-feedback loop as a Markov Decision Process", you introduce a transition probability over user preferences given actions.  Are you understanding a user's preferences to be changing over time?  Or is it just our estimate of the user's preferences that are changing over time?
- In Figure 2, does the Recommender have state that can accumulate all the virtual feedbacks v_i in each step?
- After equation (1), it says that v_i is "the embedding of user feedback to historical recommended items", but in Figure 2 it seems like v_i can only have information about the preceding recommendation and estimated reward, rather than cumulative.  Can you clarify?
- Is the reward estimator estimating something that could actually be observed?  Suppose we had a user in the loop -- can you give an example of what a reward would be for a particular set of actions?  Would it be 0/1 for whether the user clicked on one of the actions?
- In the setup for the learning task, Section 3.1 says that x_i is the "historical behaviors of user i derived from the user-item matrix X and \tilde{a}_i is the ground truth of the recommended item for the user based on his/her behavior x_i." I don't understand this.  x_i has all the items the user has clicked.  What new information is in \tilde{a}_i?  Is it referring to the a_i in the problem statement, some vector representing preferences?  Is such a vector ever known during training?  Please clarify.
- In the paragraph before equation 7, you say "given the recommended policy and the feedback generator" -- should that be "given the recommender policy"?
- In Figure 2, we see the recommender producing what seems to be a vector a_i.  Is this a set of recommendations?  Or a distribution over items from which we could sample recommendations? Or a single recommendation, in which case the graphic should change?

**Experience Assessment:**

I do not know much about this area.

**Review Assessment: Checking Correctness Of Derivations And Theory:**

N/A

**Review Assessment: Checking Correctness Of Experiments:**

I assessed the sensibility of the experiments.

**Review Assessment: Thoroughness In Paper Reading:**

I read the paper at least twice and used my best judgement in assessing the paper.

---

### Decision · Program_Chairs · 2019-12-19

**Decision:**

Reject

**Comment:**

The paper proposes to learn a "virtual user" while learning a "recommender" model, to improve the performance of the recommender system. A reinforcement learning algorithm is used for address the problem the authors defined. Multiple reviewers raised several concerns regarding its technical details including the feedback signal F, but the authors have not responded to any of the concerns raised by the reviewers. The lack of authors involvement in the discussion suggest that this paper is not at the stage to be published.